# Diffusive Steel Scrap Melting in Carbon-Saturated Hot Metal—Phenomenological Investigation at the Solid–Liquid Interface

**DOI:** 10.3390/ma12081358

**Published:** 2019-04-25

**Authors:** Florian Markus Penz, Johannes Schenk, Rainer Ammer, Gerald Klösch, Krzysztof Pastucha, Michael Reischl

**Affiliations:** 1K1-MET GmbH, Stahlstraße 14, A-4020 Linz, Austria; Johannes.Schenk@unileoben.ac.at; 2Chair of Ferrous Metallurgy, Montanuniversität Leoben, Franz-Josef-Straße 18, A-8700 Leoben, Austria; 3voestalpine Stahl GmbH, voestalpine Straße 3, A-4020 Linz, Austria; Rainer.Ammer@voestalpine.com; 4voestalpine Stahl Donawitz GmbH, Kerpelystraße 199, A-8700 Leoben, Austria; Gerald.Kloesch@voestalpine.com; 5Primetals Technologies Austria GmbH, Turmstraße 44, A-4020 Linz, Austria; Krzysztof.Pastucha@primetals.com; 6voestalpine Forschungsservicegesellschaft Donawitz GmbH, Kerpelystraße 199, A-8700 Leoben, Austria; Michael.Reischl@voestalpine.com

**Keywords:** steelmaking, basic oxygen furnace, scrap dissolution, scrap melting, thermodynamics, kinetics

## Abstract

The oxygen steelmaking process in a Linz-Donawitz (LD) converter is responsible for more than 70% of annual crude steel production. Optimization of the process control and numerical simulation of the LD converter are some of the current challenges in ferrous metallurgical research. Because of the process conditions and oxidation of impurities of the hot metal, a lot of chemical heat is generated. Therefore, steel scrap is charged as a coolant with the economical side aspect of its recycling. One of the more complex aspects is, among others, the dissolution and melting behaviour of the scrap in carbon-saturated hot metal. Heat and mass transfer act simultaneously, which has already been investigated by several researchers using different experimental approaches. The appearances at the interface between solid steel and liquid hot metal during diffusive scrap melting have been described theoretically but never investigated in detail. After an experimental investigation under natural and forced convective conditions, the samples were further investigated with optical microscopy and electron probe microanalysis (EPMA). A steep carbon concentration gradient in the liquid appeared, which started at an interface carbon concentration equal to the concentration on the solid side of the interface. Moreover, the boundary layer thickness moved towards zero, which symbolized that the boundary layer theory based on thermodynamic equilibrium was not valid. This fact was concluded through the prevailing dynamic conditions formed by natural and forced convection.

## 1. Introduction

The iron and steel industry is one of the most important global economic sectors. An increase in crude steel production up to 1670 million tons has been registered in recent years. About 72.7% of crude steel was produced by basic oxygen furnaces (BOFs), also known as Linz-Donawitz (LD) converters [1,2]. The charging materials for crude steel in a converter (besides others like slag formers) are hot metal, steel scrap, and technically pure oxygen for the oxidation of carbon, silicon, manganese, and phosphorus. Because of the exothermic reactions of oxidation, heat is generated and the temperatures increase. By using steel scrap as a coolant, the temperature can be controlled. Charging scrap also has important economic reasons, because it is an additional valuable iron source obtained through the recycling of steel. Additionally, emissions of carbon dioxide gas can be avoided by increasing the scrap ratio [1,3,4,5]. In 2017, total scrap used for steel production increased steadily to more than 600 million tons. These values show that steel scrap has become an internationally traded commodity and a worldwide symbol for the environment and recycling life cycle [2].

It is definitely desirable from the viewpoint of energy and resource improvement to utilize scrap in LD converters. In previous publications, the melting and dissolution behaviours of scrap were a common research topic conducted with the aim of increasing the scrap ratio.

A number of relevant research articles focused on the theoretical mechanisms of the different stages of scrap melting [6,7,8,9,10,11,12,13,14]. Some of them were combined with experimental investigations on the dissolution rate of scrap in liquid hot metal considering heat or mass transfer or simultaneous heat and mass transfer [12,15,16,17,18,19,20,21,22,23,24].

In recent publications, numerical modelling of LD converters and the dissolution and melting behaviour of scrap in carbon-saturated hot metal have become more interesting [4,25,26,27,28,29,30,31,32,33].

In their publications, Asai, Muchi, and Miwa [25,26,34] described theoretically—by means of a mathematical model—the LD converter operation and the mutual effect of scrap melting with process variables (e.g., melt temperature and carbon contents of scrap and melt). A more thorough theoretical description of the fundamental kinetics of the scrap dissolution process was given by Zhang and Oeters in [6] and Oeters in [7]. In [35], Szekely et al. described the heat and mass transfer phenomena involved. In [14], Glinkov et al. derived a type-III boundary condition with a slightly different approach. A more detailed theoretical description of heat and mass transfer phenomena was published by Specht and Jeschar in [13]. 

The authors of experimental studies mostly investigated the melting time of steel scraps of different sizes, shapes, and preheating temperatures. The pioneer works on steel scrap melting were carried out by Pehlke et al. in [12] and Olsson et al. [15]. Pehlke et al. studied the melting rate of cylindrical specimens in the temperature range of 1300 to 1650 °C dependent on bath agitation [12]. Olsson et al. [15] carried out experiments with rotating iron and iron–carbon cylinders and concluded that, in a range of Reynolds numbers of 350 to 10,700, the rate of dissolution was controlled by mutual counter diffusion of carbon and iron in the boundary layer [15]. Kim and Pehlke published their findings on transient heat transfer during the initial stages of steel scrap melting in [17], where shell freezing and melting were observed. In a later publication, Kim and Pehlke showed their results concerning the dissolution rates of solid pure iron into molten iron carbon alloys under isothermal conditions in [16]. In [18], Li et al. investigated the melting rate of steel bars with various sizes, shapes, and initial temperatures in a 1650 °C hot steel bath. The article was extended to a study of multi-piece scrap melting by Li and Provatas in [19]. Penz et al. investigated the dissolution behaviour of common carbon steel scrap in carbon-saturated hot metal and pointed out the differences between stagnant and dynamic bath conditions in [20]. Several authors have also reported their investigations of scrap melting and dissolution behaviour patterns in industrial experiments [36,37,38,39,40]. Experiments on a fixed reaction area were carried out by Shin et al. in [22].

As a subroutine for a dynamic model describing the LD process, a numerical solution for the scrap melting stage was published by Yorucu and Rolls in [27]. A phase-field model for the melting of scrap in the presence of convection was published by Li et al. in [28]. Their model replaced the explicit tracking of the boundary by an equation of a continuum phase field. This field described either a volume element containing only a liquid (Φ = 1) or solid phase (Φ = −1) and interpolates continuously from −1 to 1 in the transition area [28]. Theoretical aspects of scrap dissolution with simplifying assumptions were included in a Fourier series-based analytical solution by Sethi et al. in [29]. The model was critically compared with a finite difference solution, whereby the analytical model was concluded to be more accurate [29]. A number of modelling approaches were published and compared by Shukla et al in [30]. A 2D numerical simulation tool to investigate the melting process of heavy scrap pieces during LD heat was published by Guo et al. in [31]. In their conclusion, they mentioned that the cutting size of a slab had little influence on the melting time when the ratio between width and thickness was greater than 2, where the location of the scrap, relative to the oxygen jet, had a more significant influence [31]. Recent publications and research activities on creating a scrap melting model were conducted by Kruskopf et al. in [4,32,33]. Kruskopf discretized his model into a moving numerical grid and solved enthalpy and carbon concentration equations for solid material [32].

One of the main recurring assumptions in all experiments, theoretical analyses, and numerical approaches reported was the definition of the boundary carbon concentration, which was assumed to be the liquidus carbon concentration in a binary Fe–Fe_3_C phase diagram. For that reason, the authors investigated in detail the melting and dissolution behaviours of carbon steel scrap. Research findings are shown in the present paper.

## 2. Phenomenological Understanding of Scrap Melting and Dissolution Processes

Isothermal investigations into the mass transfer from solid to liquid were carried out by Lommel and Chalmers in [41]. They investigated the transport behaviour from solid lead into liquid lead–tin alloy, whereby the diffusion of tin atoms into the solid lead was not a necessary condition for mass transfer in the absence of stirring the liquid. In the studied lead/lead–tin alloy system, the initial concentration of the liquid was lower than that of the solid. Nevertheless, mass transport in the liquid adjacent to the interface occurred by volume diffusion through a boundary layer (C_HM_) of thickness δ. 

In the research work of Pehlke et al., rotating or static cylindrical rods of various sizes were submerged from room temperature into hot metal baths under different conditions and temperatures [12]. Formation of a hot metal shell was reported in the initial stages, which retarded the dissolution and lasted approximately 30 s. Gol’dfarb and Sherstov pointed out that a steel scrap ball of 20 cm in diameter had shell formation and melting that lasted approximately 2 min [10]. Nomura and Mori concluded in [11] that shell formation was negligible unless the scrap was thick. During the initial stages of immersion, heat was also transferred from the melt to the scrap until an equilibrium temperature between scrap and hot metal was reached [12]. Penz et al. showed in [42] that ultra-low carbon steel scrap cylinders 12 mm in diameter, submerged into hot metal, containing approximately 4.5 wt.-%, would reach isothermal conditions after less than 15 s immersion time. Similar results were published by Penz et al. in [21], where they defined the heat transfer coefficient between the steel scrap and the hot metal in the initial stages of immersion. Olsson et al. only reported on mass transfer measurements after reaching equilibrium temperature between scrap and hot metal [15]. Penz et al. defined in [20] the ablation rate of a cylindrical specimen and mass transfer coefficient by using a diffusive scrap melting formula published by Zhang and Oeters in [6]. Moreover, Li et al. in [18,19] and Xi et al. in [23] reported experiments on scrap melting in liquid iron–carbon melts at temperatures above the liquidus temperature of iron. To summarize all these findings, it is clearly shown that scrap melting is a complex mechanism combining simultaneous heat and mass transfers. The following is a proposal on how to divide the scrap melting process into three stages:Stage 1: Solidification of a liquid hot metal layer on the surface of cold scrap, which will re-melt fast after enough superheat is available. Heat and mass transfer work simultaneously.Stage 2: Dissolution of the scrap, depending on the carbon composition of the hot metal and the scrap, also defined as diffusive melting. At this stage, superheat will be consumed for promoting necessary mass transfer. Because heat transfer is much faster than mass transfer, and the carbon content in the solid steel is much lower than in the liquid melt, only mass transfer has to be considered.Stage 3: A forced or convective scrap melting stage will be reached if the temperature of the hot metal exceeds the melting temperature of the scrap. In this case, only heat transfer should be considered.

To sum up the phenomenological understanding, it is necessary to highlight that mass transfer—especially from carbon—is important to investigate. Further, previous publications have shown that bath agitation and temperature influence the dissolution and melting behaviour of scrap. In Figure 1, the phenomena of heat and mass transport at the solid–liquid interface are depicted. The subscript init stands for the initial concentration or temperature of the scrap, which is uniform at time t = 0 over the whole transverse section. The initial temperature and carbon concentration of the hot metal are T_HM_ and C_HM_, respectively. Temperature and concentration gradients will occur in both the liquid and solid. At the interface, a slightly lower temperature T_S_ than the hot metal will be reached. The equilibrium concentration achieved in the liquid for the temperature T_S_ is $C_{l}^{*}$ and $C_{s}^{*}$ in the solid. The boundary layers in the liquid for heat and mass transfer are $\delta_{T}$ and $\delta_{C}$, respectively. According to Goldberg and Belton in [43] and Bester and Lange in [44], the diffusion coefficients of carbon in liquid iron and iron–carbon alloys are significantly higher than in solids, which could diminish the concentration boundary layer in the liquid dramatically.

## 3. Theoretical Description

In the present article, the authors focus on diffusive scrap melting under isothermal conditions. Shurygin and Shantarin proposed in [8] that the rate of dissolution of an iron disk in hot metal was limited by the diffusion of elements in the boundary layer. The expression for the boundary layer thickness $\delta_{C}$ they used is given by Levič in [45]. The proposed rate of dissolution, N, including Levič’s expression, is given in Equation (1) [8,15,45].
(1)$$N=\frac{D_{s}}{\delta_{C}}A\left(C_{scrap}-C_{HM}\right)=1.95*r²{D_{s}}^{2/3}\nu^{-1/6}\omega^{1/2}\left(C_{scrap}-C_{HM}\right).$$

In Equation (1), $D_{s}$ is the diffusion coefficient in the solid, *A* is the surface area and *r* defines the radius of the sample. The parameters *ν* and *ω* describe the kinematic viscosity and the angular velocity. Nevertheless, Olsson et al. concluded in [15] that the expression proposed by Shurygin and Shantarin would lead to high diffusivity values. Olsson et al. noted that the liquid and solid at the interface were in an equilibrium relation. They assumed that for that reason, the liquid at the interface had liquidus composition corresponding to the bath temperature $\left(C_{liq}\right)$, which is given in Equation (2). These findings were based on the considerations published by Lommel and Chalmers in [41]. Equation (2) is only valid if equal densities in the solid and liquid are considered [15,41].
(2)$$-\frac{dr}{dt}=k_{met}*ln\left(1+\left(\frac{\%C_{liq}-\%C_{HM}}{\%C_{Scrap}-\%C_{liq}}\right)\right).$$

In Equation (2), $k_{met}=D/\delta$ is the mass transfer coefficient (m/s), which is the ratio between the diffusion coefficient D and the boundary layer thickness δ. 

Assuming that solid steel scrap has a more uneven composition than liquid hot metal, as usual in an LD converter, the melting is extended with dissolution phenomena, including multiphase systems with temperature as well as chemical composition-dependent liquidus and solidus lines. Based on Figure 2, which shows the Fe–Fe_3_C phase diagram of common S235JR construction steel scrap, the solidus and liquidus lines could be estimated. The square blue points in Figure 2 show the specific isothermal carbon concentrations of the scrap ($C_{scrap}$) and an assumed liquid hot metal composition $\left(C_{HM}\right)$ as well as the solidus ($C_{s}^{*}$) and liquidus ($C_{l}^{*})$ concentrations at 1300 °C. In [9], Krupennikov and Filimonov expressed the direct solution of steel in iron carbon melts in two successive stages: the transition of iron atoms through the phase boundary (kinetic stage) and the subsequent diffusion through the boundary layer into the melt (diffusion stage). The result of their theoretical consideration was that the carbon concentration at the surface of the solid had to be equal to $C_{s}^{*}$ at the solidus temperature. This led to a chemical potential difference $\Delta \mu \left(C_{s}^{*},C_{l}^{*}\right)$ between the iron atoms on the surface of the solid and in the melt at the phase boundary. Because of the breakdown of the solid’s crystalline structure, which results in a reduction of activation energy characterizing the kinetic stage, a liquid phase will form in the decomposition of a supersaturated solid. However, they mentioned that no significant kinetic factors retarding the melting process had been observed yet [9]. As the authors presented in [46] with their dynamic LD converter model, and Asai and Muchi [25] showed based on the investigations of Oya in [47], the carbon concentration of the hot metal could also be just below the liquidus line in a binary phase diagram, which led to a negative logarithmic term in Equation (2). In fact, a negative logarithmic term results in an increase in scrap during the blowing period of an LD converter.

The mass balance on the surface according to Figure 1 and the theoretical description of Specht and Jeschar in [13] is given in Equation (3), where $k_{met}^{\prime }$ is the mass transfer coefficient in m/s, $D_{s}$ is the diffusivity into the solid in m²/s, and $\rho_{HM}$ and $\rho_{scrap}$ are the densities of the hot metal and scrap in kg/m³, respectively.
(3)$$k_{met}^{\prime }*\rho_{HM}*\left(C_{HM}-C_{l}^{*}\right)=-\rho_{scrap}*D_{s}*{\frac{dC}{dx}}_{|x=r\left(x,t\right)}-\rho_{HM}*\left(C_{l}^{*}-C_{s}^{*}\right)\frac{dr\left(x,t\right)}{dt}.$$

Glinkov et al. presented a slightly different aspect in [14]. In their theoretical exploration, they determined that the interface composition was equal in the solid and the liquid ($C_{int}$). It was not clearly defined if the interface carbon concentration in Glinkov’s research work was either $C_{s}^{*}$ or $C_{l}^{*}$.

From Figure 1 it is known that carbon is transported steadily to the interface through convection and will diffuse into the solid scrap. The fundamental equation for diffusion is Fick’s second law, which is given in Equation (4) with the three-dimensional differential operator ∇ [48,49,50,51,52].
(4)$$\frac{\partial C}{\partial t}=\nabla \cdot \left(D\nabla C\right).$$

The general equation can be transferred to the more common form in Equation (5), where the axial flow in the *x*-direction is given with the boundary conditions ∂/∂z=∂/∂y=0 and $0<r\left(x,t\right)<\infty$ according to the sketch in Figure 1: (5)$$\frac{\partial C}{\partial t}=\nabla \cdot \left(D\nabla C\right),$$
where *C* is the concentration and the diffusion coefficient is $D_{s}$. Assuming that the state of constant ambience concentration ($C_{HM}$) is given, the melting rate can be stated as constant.
(6)$$v=-\frac{dx}{dt}=-\frac{dr\left(x,t\right)}{dt}=const.$$

The boundary conditions for the mass transfer are defined in [13] and according to the sketch in Figure 1 as follows:(7)$$C\left(x=0\right)=C_{scrap}\;\mathrm{and}\;C\left(x=\infty \right)=C_{HM}.$$

Kim and Pehlke reported in [16] a carbonised layer of 58 µm at 1242 °C in stagnant melt conditions. Experiments with a fixed reaction area were carried out by Shin et al. in [22] and Nomura and Mori in [11]. Shin et al. approximated the carbonised layer thickness to be 114, 108, and 57 µm at 1230, 1325, and 1415 °C, respectively, by using temperature- and carbon-dependent extrapolated diffusion coefficients [22]. Nomura and Mori reported an estimated diffusion boundary layer thickness of 50 to 100 µm at decreasing temperatures in the range of 1420 to 1200 °C, respectively [11]. According to those measurements, the boundary condition $C\left(x=0\right)=C_{scrap}$ could be assumed for samples with a thickness >1 mm. With the boundary conditions and the constant melting rate, the solution to Equation (5) is expressed in Equation (8).
(8)$$\frac{C-C_{scrap}}{C_{s}^{*}-C_{scrap}}=\exp\left(-\frac{v}{D_{s}}*X\right).$$

The surface gradient will be as given in Equation (9) if the interface concentration in the solid is assumed to be the solidus concentration at the presumed temperature.
(9)$${\frac{dC}{dx}}_{|x=r\left(x,t\right)}=-\left(C_{s}^{*}-C_{scrap}\right)\frac{v}{D_{s}}.$$

As long as the widely spread consideration in literature is used, that the densities of the scrap $\rho_{scrap}$ and hot metal $\rho_{HM}$ are equal, the mass balance of Equation (3) can be rewritten and simplified to Equation (10).
(10)$$-\frac{dr}{dt}=k_{met}^{\prime }*\left(\frac{C_{HM}-C_{l}^{*}}{C_{l}^{*}-C_{scrap}}\right).$$

According to Zhang and Oeters in [6], the mass transfer coefficient $k_{met}^{\prime }$ used in Equation (10) is connected to $k_{met}$ by the function expressed in Equation (11) using the dimensionless value ξ [6]. It is the ratio of the velocity of the boundary movement to the mass transfer.
(11)$$f\left(\xi \right)=\xi /\left(1-e^{-\xi }\right).$$

With high values of ξ, the mass transfer will increase, which means that the mass transfer is only dependent on the velocity of the boundary movement. A detailed description of this explanation is given by Zhang and Oeters in [6]. Using Zhang and Oeters’ expression in Equation (11) with the mass balance of Equation (10), the expression of the mass balance will result in the initial Equation (2). 

From a theoretical point of view, the dissolution rates are controlled by the interdiffusion of iron and carbon in the liquid boundary layer. Therefore, the kinetics of scrap dissolution essentially belong to the class of moving boundary problems with phase changes, which is also known as a Stefan problem. A more comprehensive theoretical description of Stefan problems can be found in various research articles (e.g., [53,54,55]). It should be kept in mind that the values of mass transfer will also depend on scrap geometry, the condition of the liquid bath, and various physical parameters that include the diffusion coefficients.

In a literature research process, two papers investigated directly the surfaces of scrap samples. In [24], Sun et al. presented their experimental observations of spherical scrap melting in hot metal. Spherical scrap with a mass of 0.23 to 0.25 kg fixed on a molybdenum rod was immersed into a 70 kg hot metal bath with temperatures between 1300 and 1600 °C. Electron probe microanalyses (EPMAs) were carried out on the surface of specimens. They detected a slightly higher amount of carbon counts close to the surface, concluding that it might be insignificant. Xi et al. carried out thermal simulation experiments with GCr15-bearing steel samples at temperatures of 1500 °C and 1600 °C, and they published their results in [23]. With optical microscopy analyses, a sharp interface between the frozen hot metal shell (primary carbide) and the mother scrap was detected. In the inner layers of the steel bar, the microstructure changed to martensite due to water quenching. Xi et al. concluded that it was reasonable to assume that the melting process was less affected by mass transfer of carbon, and that the melting process was mainly controlled by heat transfer at the investigated temperatures [23].

## 4. Experimental Investigation

### 4.1. Experimental Setup and Melting

In a previous study, the authors submerged S235JR construction steel in carbon-saturated hot metal with a carbon concentration of approximately 4.58 wt.-% [20]. The primary objective of Penz et al. was to determine the mass transfer coefficient under stagnant or turbulent bath conditions and various temperatures. To simulate turbulent bath conditions, the samples were rotated with a rotation speed of 100 rpm. With the geometric and physical parameters of the experimental setup, the theory of Taylor–Couette flow was used to define when laminar flow became unstable and more turbulent [20,56,57]. The initial chemical compositions and experimental data are listed in Table 1. Thereby, a mass of 330 g of hot metal was heated in an aluminum oxide crucible in a Carbolite Gero high temperature vertical tube furnace. The heating rate was specified at 300 K/min. For oxidation prevention, the furnace was flushed with nitrogen gas. Cylindrical specimens 12 mm in diameter were submerged into the hot metal. The initial temperatures of the hot metal were measured to be 1305, 1370, and 1450 °C. The initial temperature of the specimen was 25 °C before submerging. The submerged length was 20 mm. Through further experiments, published by the authors in [42], it was verified that an equilibrium temperature between the hot metal and the core of the scrap was reached after 10 s. The equilibrium temperatures were 1230, 1300, and 1385 °C, respectively, and are listed with the initial temperatures of the scrap and the hot metal in Table 1 [20].

### 4.2. Sample Preparation

After a specified submerging time, the samples were withdrawn and quenched with water to avoid oxidation and stop carbon diffusion immediately. To prevent any influence of the microstructure and the chemical distribution of the elements inside the samples, they were cut longitudinally with wire electrical discharge machining (wire EDM). For the cutting, deionized water was used as a dielectric. Afterwards, the sectional surface of the sample was ground with SiC sand paper and polished to 1 µm. The final polishing to 0.1 µm for electron microprobe analysis (EPMA) was carried out with an alumina suspension. EPMA was done with a JEOL JXA-8230 using a tungsten filament for generating the electron beam and five wavelength-dispersive X-ray spectroscopy (WDS) detectors. The elements investigated were carbon, silicon, manganese and phosphorus. Additionally, chromium and oxygen were partially investigated. For the latter, quantification was not possible because there was a lack of standards available. The analyzed area had a size of 240 × 2000 µm, and one detection point occupied an area of 3 × 3 µm. Measurements were carried out with an acceleration voltage of 15 kV and a probe current of 580 nA. After the EPMA, the samples were etched with nitric acid and analyzed by means of optical microscopy.

## 5. Results and Discussion

### 5.1. Effect of Bath Temperature on Dissolution

As mentioned above, the authors investigated in [20] the dissolution and melting behaviours of the scrap. For each specified immersion time, three samples were submerged into fresh, hot metal. The average rate of the radius after immersion (r(t)) with the initial radius (r0) was further determined through the weight. The detailed description of the experiments and the evaluation of the ablation rate and mass transfer coefficients can be taken from [20]. In Figure 3, the average rate of the radius after the specified immersion time with the initial radius is shown for a stagnant system. In the initial 10 s, shell freezing occurred, which reached its maximum point after approximately 5 s. The maximum radius of the frozen shell was temperature-dependent and was in the range of 15% at an initial temperature of 1450 °C to 25% at 1305 °C of the initial scrap radius. After melting of the shell, a constant dissolution of the mother scrap occurred, which was faster with increasing temperature. The light areas in Figure 3 show the standard deviation of the determined radii. The results of the average dissolution rate of the radius after immersion (r(t)) with the initial radius (r0) are published in [20].

### 5.2. Optical Microscopy Observations

To summarize the observations of optical microscopy, a few samples were taken for this publication. Two of the samples (162 and 107) presented in detail were submerged—without rotation of the sample—into the hot metal to receive stagnant conditions influenced only by natural convection. The Nital-etched longitudinal area of samples 162 and 107, which were submerged for 20 s at an initial temperature of 1305 °C and 180 s at 1370 °C, respectively, were visible in the mosaic picture in Figure 4 and Figure 5. Through the energy balance, the equilibrium temperature between scrap core and hot metal dropped to 1230 or 1300 °C, respectively. Sample number 59 was submerged into the hot metal for 60 s at an initial temperature of 1450 °C and a rotational speed of 100 rpm. Owing to this speed, turbulent conditions occurred in the hot metal. The Nital-etched longitudinal area of sample number 59 is shown in Figure 6.

In Figure 4, Figure 5 and Figure 6, the submerging direction was in an axial direction (z). It can be seen that through the removal from the melt, a droplet of hot metal stuck to the ground surface of the scrap on the left side of the figures. On the surface of the sample in Figure 4, residuals of the frozen shell were still visible. The scrap itself melted stronger in the radial direction (r), which could be explained by local density differences according to the temperature gradient in the boundary layer of the hot metal. With increasing immersion time, dissolution also started on the ground surface. A small bright layer was visible in the mother scrap, which was an indication of a chemical event during dissolution. 

After the EPMA measurements, the same section was investigated by means of optical microscopy. Between the two steps of investigation, only Nital etching was carried out. No abrasive steps, like polishing or grinding, were necessary. For this reason, the tracks of the EPMA were barely visible, and a comparison could be made with the outputs of the EPMA. 

In Figure 7, the section of sample 162 is shown where the EPMA was carried out at 100× magnification. The rest of the frozen shell of primary carbide is on the left side of the picture. As a result of the measurements carried out by the authors in [42], it was known that the temperature in the scrap reached the austenite temperature and would form martensite in the subsequent water-cooling process. Between those areas, a significant layer more than 100 µm in thickness could be seen, which was identified in Figure 4 as the bright area.

The same microstructure was also found in sample 107 at an equilibrium temperature between scrap core and hot metal of 1300 °C, shown in Figure 8 at 100× magnification. In this picture, it can be clearly seen that there was a sharp interface between the primary carbide and the mother scrap. The bright area was definitely a part of the scrap. In the primary carbide, small residual grains of the scrap were also visible, which resulted from an uneven dissolution through the grain boundary diffusion of carbon.

In Figure 9 the section of EPMA of sample 59 is shown at 200× magnification. The sharp interface between solid scrap and residual primary carbide as well as the diffusion of carbon between the austenite grains was clearly visible.

In all figures, it was noticeable that the primary carbide was in direct contact with the interface during the dissolution process. Scattered former grains of the scrap, liberated from the scrap, were still close to the dissolution interface, which was an indication of a low transportation speed of solid particles in the hot metal.

### 5.3. Electron Probe Microanalysis (EPMA) Observations

The EPMA measurements were carried out in the certified and accredited laboratory of voestalpine Forschungsservicegesellschaft Donawitz GmbH/Section Material Analytics. In the following figures, the measured fields on the surface of the scrap sample are shown. Through rapid removal of the sample from the hot metal and subsequent water cooling, residual melt stuck to the surface and solidified immediately. It was possible to measure with EPMA the element distribution crossing the interface between the mother scrap and the residual primary carbide. In all figures, the lower graphs show the average value of the 80 measurement points of an area 3 µm × 3 µm each in *y*-distance in the colored shape image. The legends in all figures are on the same scale to compare them to each other.

The colored shape image in Figure 10 shows the carbon distribution of sample 162. The red area on the left side shows the frozen residual melt with a carbon composition of more than 4 wt.-%, including liberated grains from the scrap with a composition below the solidus composition. To compare with the microstructure picture in Figure 7, the red area stopped at the brown sharp interface. From the average carbon composition in the graph, it can be seen that carbon from the hot metal diffused approximately 100 µm into the scrap. A steep gradient from the solidus point at the interface to higher carbon contents in the former melt region was visible. Due to distortions of the liberated grains, it is not clearly readable from the average graph that the carbon composition on the liquid side of the interface steps from the interface carbon composition directly to the hot metal carbon composition. It could be seen that the average silicon formed a layer on the scrap side surface with silicon contents four times higher than in the scrap and one and a half times higher than in the hot metal. Inside the scrap, it was visible that the small silicon peaks were congruent with the oxygen counts, which was a sign of silicon oxide inclusions.

In Figure 11, a comparison of the average carbon diffusion profiles of samples with different immersion times at an equilibrium temperature of 1230 °C and natural convection conditions in the bath is shown. The detected lines were cut at the detected interface between the solid and liquid melt. It can be seen that the diffusion profile was nearly equal, and the diffusion depth was approximately 200 µm, except at the two shortest immersion times of 10 and 20 s where the profile was not well developed. At an immersion time of 240 s, carbon diffusion was much deeper, which was a result from the measuring position being too close to the ground surface of the sample. The same behaviour and approximately the same diffusion depth of the carbon were detected for samples with a rotation speed of 100 rpm so that turbulent conditions existed in the melt. The comparison of the average carbon compositions from starting at the liquid–solid interface for turbulent bath conditions is shown in Figure 12. The received interface carbon content in the solid $\left(C_{interface}\right)$ was equal to the solidus carbon content at this temperature.

The colored shape image in Figure 13 shows the carbon distribution of sample 107, which was immersed for 180 s. The adjusted equilibrium temperature between scrap core and hot metal was 1300 °C. Like in Figure 10, again a sharp interface can be detected between the scrap and hot metal. In the solid area, the carbon concentration at the interface was—with approximately 1.5 wt.-% carbon—slightly higher than the solidus composition. It seemed that mass transport and dissolution were partly inhibited from the silicon layer formed in the scrap close to the surface. To compare with the microstructure picture in Figure 8, the red area stopped at the bright white, sharp interface. A few liberated grains were also visible on the left side and former liquid area. These liberated grains again led to distortions in the average carbon composition. In detail, the contour levels of the interface carbon composition in the solid, the carbon concentration of the liquidus line, and the carbon concentration of the hot metal were identical. This fact showed that there was a steep concentration gradient in the liquid and a boundary layer in the liquid several times smaller than the carbon boundary layer in the solid phase. These results showed that the boundary layer in the liquid moved towards zero in comparison to the carbon concentration boundary layer in the solid. Ultimately, there was no equilibrium between the carbon concentration of the liquidus line and the carbon concentration at the interface on the solid side. The authors concluded from those facts that the dissolution process was not describable in the liquid using Fick’s boundary layer theory. It was apparent that the strong dynamic conditions, caused by natural or forced convection, influenced the dissolution process. Additionally, possible local phenomena influencing the dissolution process may be Rayleigh–Bénard convection, which is based on local density differences, or Marangoni convections, based on local surface tensions. A conspicuous characteristic was also the enrichment of silicon in the solid scrap close to the interface.

Figure 14 and Figure 15 depict the average carbon diffusion profiles for stagnant conditions, where only natural convection arises, and turbulent conditions of the hot metal, respectively. The measured values from various samples with specific immersion times at an equilibrium temperature between scrap core and hot metal of 1300 °C started again from the detected solid–liquid interface. As already mentioned before, at 1230 °C, an approximately equal carbon diffusion profile in the solid arose for all immersion times. The thickness of the carburized layer was between 150 and 200 µm, which was more than double compared to previous publications. It is mentionable that diffusion depth is a little bit deeper for conditions with natural convection. This resulted from a lower bath agitation and, therefore, also a lower dissolution rate. The carbon concentration at the interface ($C_{interface}$) was approximately 1.5 wt.-%.

The colored shape image in Figure 16 shows the carbon distribution of sample 59, which was immersed for 60 s in hot metal with turbulent bath conditions realized through a rotational speed of the sample of 100 rpm. The adjusted equilibrium temperature between scrap core and hot metal reached 1385 °C. Again, there was a steep gradient between the carbon concentration at the interface ($C_{interface}$) up to the hot metal carbon concentration of more than 4 wt.-% carbon. The average carbon concentration in the graph below was influenced by liberated grains. In comparison to the microstructure in Figure 9, it was observable that the solidus concentration was deep inside the scrap, and the dissolution was again inhibited. A further point to mention is that there was no possibility to clearly define in which state (e.g., graphite or as a component) the dissolved carbon existed in the boundary layer region. As at lower temperatures the contour levels of the interface carbon composition in the solid, the carbon concentration of the liquidus line, and the carbon concentration of the hot metal were identical.

The average carbon diffusion profiles for stagnant or turbulent conditions at an equilibrium temperature of 1385 °C showed similar results to those at lower temperatures. The carbon concentration at the interface ($C_{interface}$) was approximately 1.6 wt.-%. 

To emphasize silicon enrichment, Figure 17 shows the EPMA-measured silicon distribution of sample 59, parallelly measured with the carbon distribution from Figure 16. It was clearly visible that a silicon-enriched layer was formed in the solid scrap at 125 µm < *x* < 140 µm from the measurement starting point. The first former liquid regions, detected by carbon distribution, arose at *x* < 125 µm. Such an enrichment was observed in all samples during the entire investigation period. In the literature, there has not been any similar behaviour reported thus far. A theory regarding this phenomenon is that the dispersed silicon oxides in the mother scrap will stay at the lowest energy level, which is at the interface. They will be reduced by the transferred carbon, which may be a retarding factor of the dissolution process. Additionally, a possible occurrence of diffusive events at the reported temperature levels, for example mass transfer of silicon, cannot be ruled out in this case. A closer investigation into this phenomenon has to be carried out in future research and experimental investigations.

### 5.4. Discussion of Mass Balance According to EPMA Investigations

From the results of the EPMA measurements, it was evident that the carbon concentration on the solid side of the solid–liquid interface increased through diffusion of carbon to an interface carbon concentration ($C_{interface}$). For an equilibrium temperature between scrap core and hot metal of 1230 °C, the interface carbon concentration was defined by $C_{interface}=C_{s}^{*}$. With increasing temperature, the interface carbon concentration increased to values of 1.5 wt.-% carbon at 1300 °C and 1.6 wt.-% carbon at 1385 °C. In line with the phase diagram in Figure 2 of the scrap composition determined, these concentrations were in the two-phase area. In the liquid region, a steep gradient from the interface carbon concentration to the carbon composition of the hot metal was clearly recognized. Theoretical analyses reported in former publications always expected an equilibrium system, which was explained by means of the boundary layer theory and Fick’s second law. In the case of scrap dissolution, the present paper shows with the EPMA measurements that the boundary layer in the liquid went to zero in comparison with the diffusion boundary layer inside the solid scrap. Based on the measurements, the authors will amend the mass balance of Equation (3) to the following expression given in Equation (12):(12)$$k_{met}^{\prime }*\rho_{HM}*\left(C_{HM}-C_{interface}\right)=-\rho_{scrap}*D_{s}*{\frac{dC}{dx}}_{|x=r\left(x,t\right)}.$$

The term on the left side of Equation (12) expresses the carbon transport in the liquid melt with the carbon concentration difference. According to the evaluated results from the EPMA, the difference is between the hot metal composition $C_{HM}$ and an interface carbon concentration applicable for $C_{l}^{*}>C_{interface}\geq C_{s}^{*}$. To close the mass balance, only the expression of carbon diffusion into the solid scrap will be in the counter direction. This fact is disputed through non-equilibrium and heavy dynamic conditions occurring under both natural and forced convection in the liquid. By the use of Equation (9) and the ratio of the velocity of the boundary movement of Zhang and Oeters in Equation (11), the mass balance can be transformed into the following expression, which will describe the diffusive dissolution behaviour of steel scrap in liquid hot metal. It is valid for the following temperature-dependent expression $C_{l}^{*}>C_{interface}\geq C_{s}^{*}$:(13)$$-\frac{dr}{dt}=k_{met}*ln\left(\left(\frac{\left(\%C_{HM}-\%C_{interface}\right)*\rho_{HM}}{\left(\%C_{interface}-\%C_{scrap}\right)*\rho_{scrap}}\right)+1\right).$$

Additionally, it has to be mentioned that the dissolution behaviour will also be influenced through other elements, which is reasonable due to the observed enrichment of silicon. Furthermore, an influence through local phenomena (e.g., Rayleigh-Bénard convection) might be given and should definitely be investigated in detail.

## 6. Conclusions

Scrap melting and dissolution is indeed a complex and complicated process including simultaneous heat and mass transfer. The present study describes the analytic determination of common steel scrap dissolution tests in liquid carbon-saturated hot metal. In previous publications, a thermodynamic equilibrium between the liquid metal and the solid scrap was assumed, allowing the steel scrap dissolution to be explained using Fick’s second law and by means of a binary Fe–Fe_3_C phase diagram of the used scrap. 

As a result of systematic approaches, the scrap samples immersed into the hot metal were removed and subsequently water-cooled. Without additional heat input, the sample preparation was carried out to perform a series of electron probe microanalysis (EPMAs) on the scrap surface and the residual melt, which could not flow away rapidly enough before quenching. By comparison with optical microscope measurements, the area analyzed using EPMA could have been used to provide further information.

It was observed that in the liquid area, a steep concentration gradient of carbon existed, which started at an interface carbon concentration equal to the interface carbon concentration in the still solid material. Through maximum increase in the concentration gradient in the liquid, ending at the hot metal carbon concentration, it was concluded that the boundary layer in the liquid moved towards zero, and no distinctive boundary layer was formed. Notably, strong dynamic conditions, caused by natural or forced convection, influence the dissolution process. The appearance of local convection phenomena influencing the dissolution process (e.g., Rayleigh-Bénard convection) cannot be excluded. Through the detected interface carbon concentration, a new approach for mass balance to describe the dissolution process was outlined. Moreover, it was visualized that the diffusion of carbon into the solid was constant at every investigated immersion time step, which was an indication of a constant dissolution rate under isothermal conditions. The carburized boundary layer thickness in the solid material reached between 150 to 200 µm, which was approximately two times more than that reported in previous publications. The new approach reported can be easily included into dynamic LD-converter process models, which are based on thermodynamic considerations. As long as the equilibrium temperature of the liquid hot metal and the scrap is below the two-phase melting stage at the initial carbon concentration of the scrap, the equation given in Equation (13) might be used. Further, it is necessary that the process model implies an approximation of the mass transfer coefficient as well as an estimation of the interface carbon concentration according to the rule $C_{l}^{*}>C_{interface}\geq C_{s}^{*}$. 

Conspicuous characteristic silicon enrichments were detected in the solid scrap close to the solid–liquid interface. Due to the high temperatures and large amounts of silicon oxides in the scrap, it was concluded that the oxides will be trapped at the interface and reduced through the carbon. It is absolutely necessary to investigate in further research the behaviour of other elements besides carbon on their diffusivity into the solid steel and mutual interference with the carbon mass transfer.

In summary, the outcomes of the present study clearly indicate that more individual and adequate experiments will be necessary in future research work to describe steel scrap melting and dissolution behaviours in hot metal.

## Figures and Tables

**Figure 1 materials-12-01358-f001:**
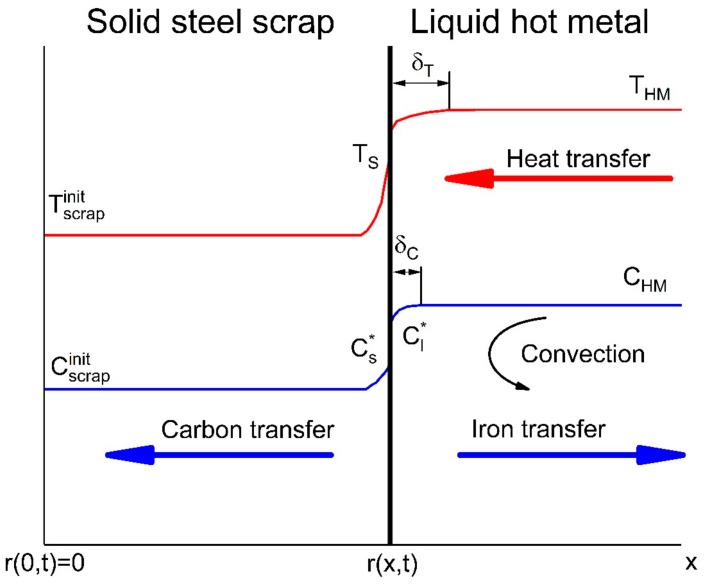
Schematic diagram of mass and heat transfer between cold scrap and liquid hot metal.

**Figure 2 materials-12-01358-f002:**
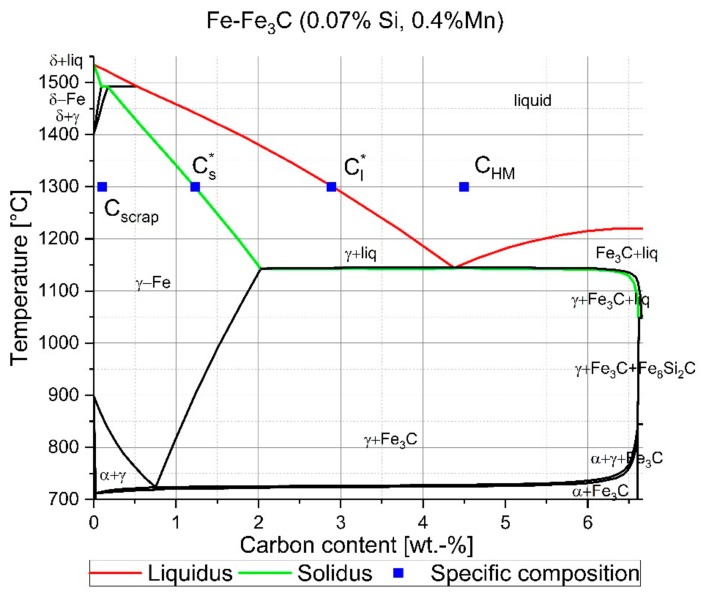
Fe–Fe_3_C phase diagram of common S235JR construction steel scrap including schematic points of carbon concentrations.

**Figure 3 materials-12-01358-f003:**
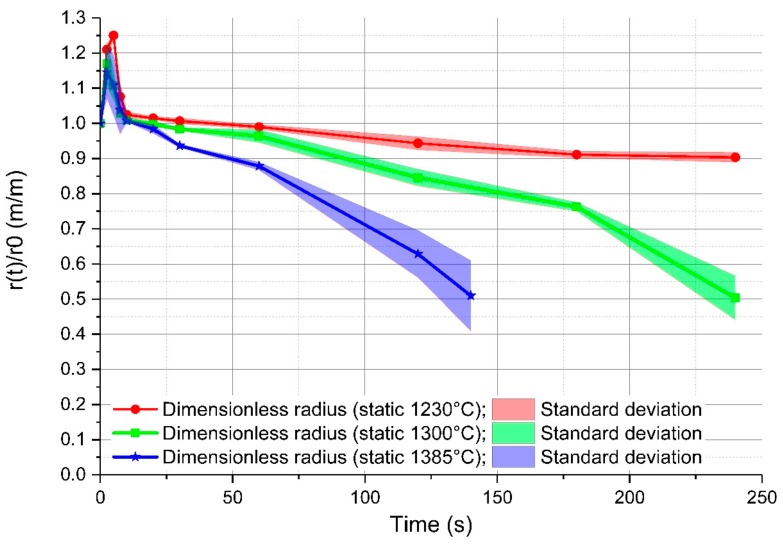
Effect of bath temperature on the melting rate of cylindrical scrap samples in hot metal.

**Figure 4 materials-12-01358-f004:**
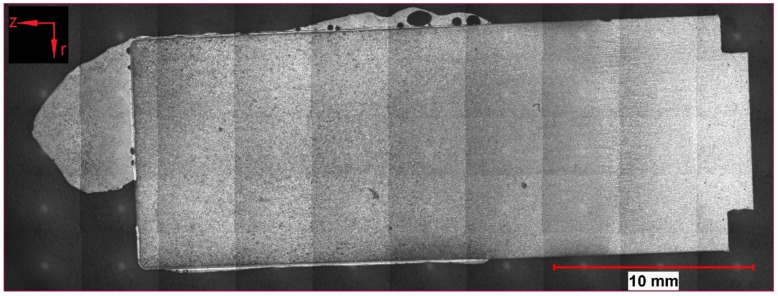
Nital-etched longitudinal area of stagnant sample number 162, submerged for 20 s into hot metal with an equilibrium temperature between scrap core and hot metal of 1230 °C.

**Figure 5 materials-12-01358-f005:**
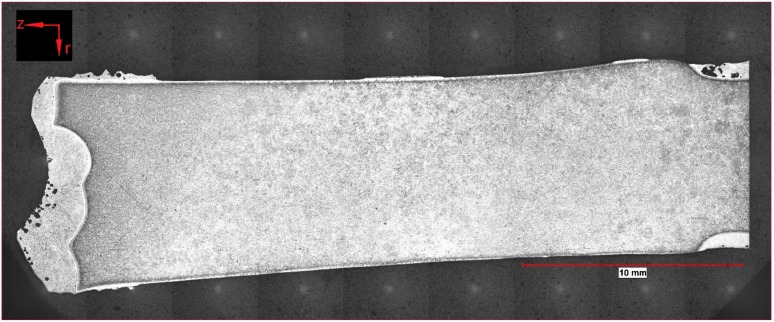
Nital-etched longitudinal area of stagnant sample number 107, submerged for 180 s into hot metal with an equilibrium temperature between scrap core and hot metal of 1300 °C.

**Figure 6 materials-12-01358-f006:**
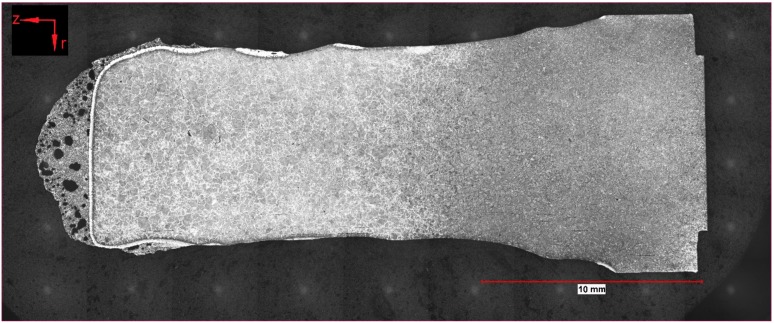
Nital-etched longitudinal area of rotating sample number 59, submerged for 60 s into hot metal with an equilibrium temperature between scrap core and hot metal of 1385 °C.

**Figure 7 materials-12-01358-f007:**
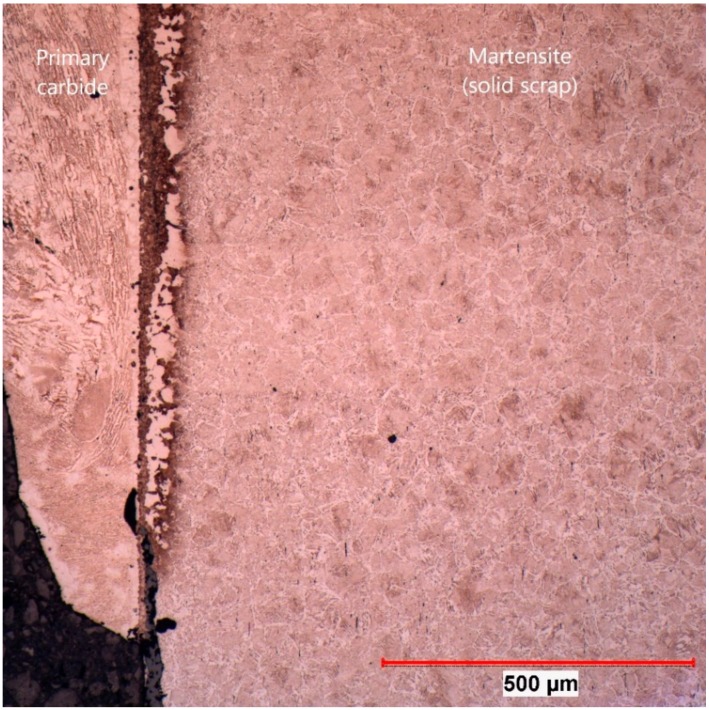
Section where the electron probe microanalysis (EPMA) measurement was carried out for sample 162 (20 s immersion time at 1230 °C equilibrium temperature).

**Figure 8 materials-12-01358-f008:**
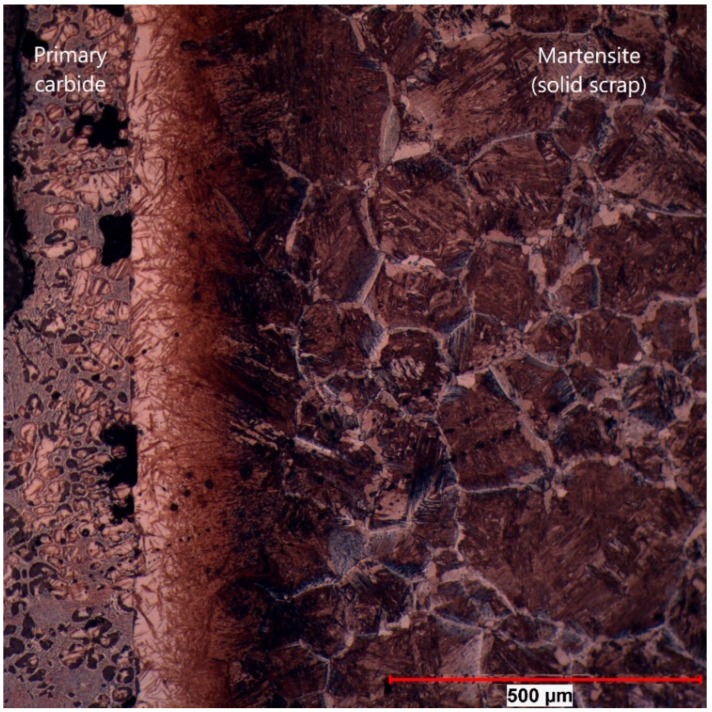
Section where the EPMA measurement was carried out for sample 107 (180 s immersion time at 1300 °C equilibrium temperature).

**Figure 9 materials-12-01358-f009:**
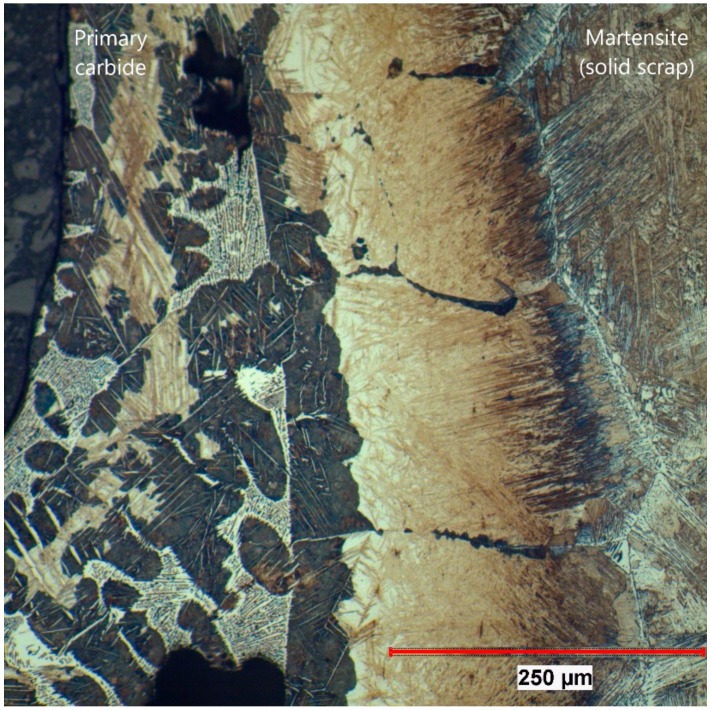
Section where the EPMA measurement was carried out for sample 59 (60 s immersion time at 1385 °C equilibrium temperature).

**Figure 10 materials-12-01358-f010:**
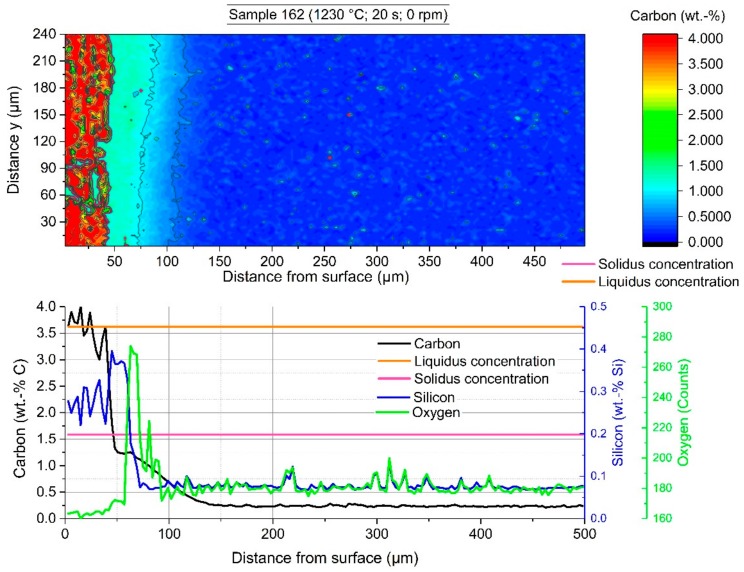
EPMA-measured carbon distribution of sample 162 with an immersion time of 20 s at 1230 °C equilibrium temperature.

**Figure 11 materials-12-01358-f011:**
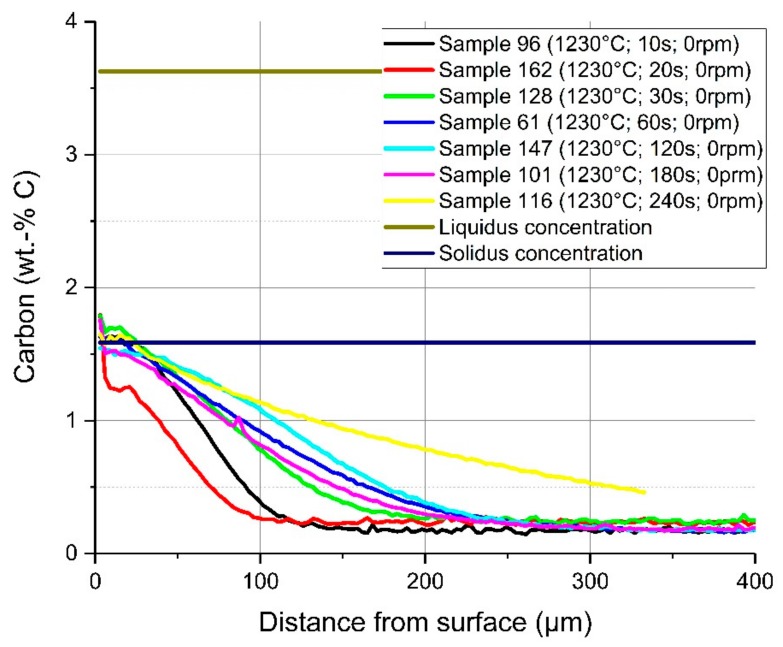
Average carbon composition and diffusion depth from the detected solid–liquid interface under natural convection in the melt at 1230 °C equilibrium temperature.

**Figure 12 materials-12-01358-f012:**
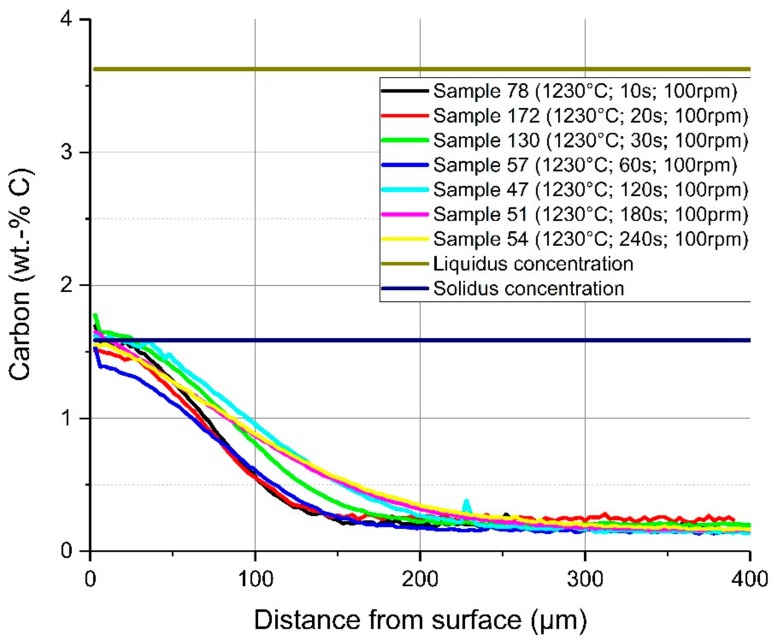
Average carbon composition and diffusion depth from the detected solid–liquid interface under forced convection in the melt through rotation of the sample at 1230 °C equilibrium temperature.

**Figure 13 materials-12-01358-f013:**
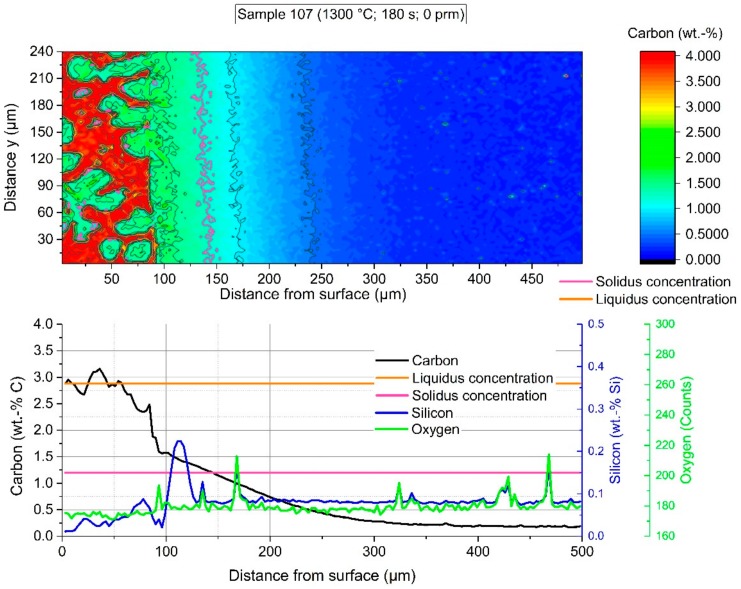
EPMA-measured carbon distribution of sample 107 with an immersion time of 180 s at 1300 °C equilibrium temperature.

**Figure 14 materials-12-01358-f014:**
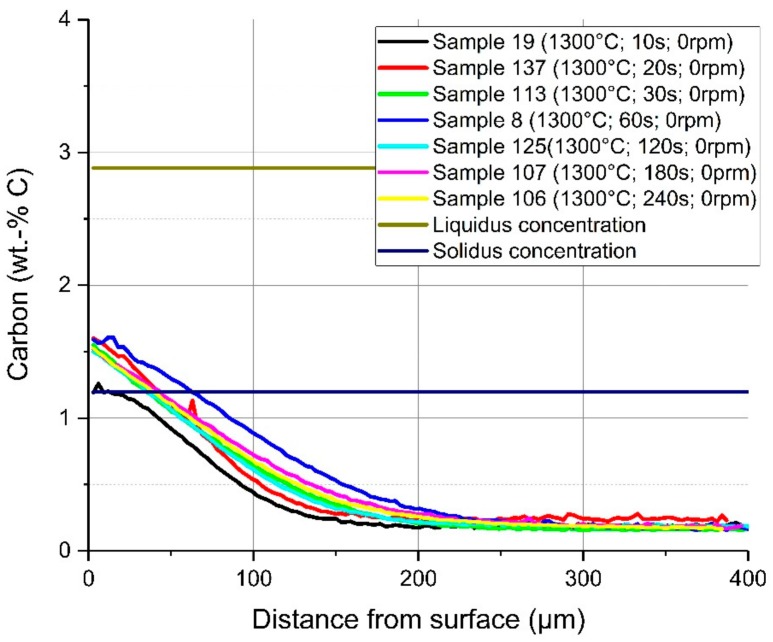
Average carbon composition and diffusion depth from the detected solid–liquid interface under natural convection in the melt at 1300 °C equilibrium temperature.

**Figure 15 materials-12-01358-f015:**
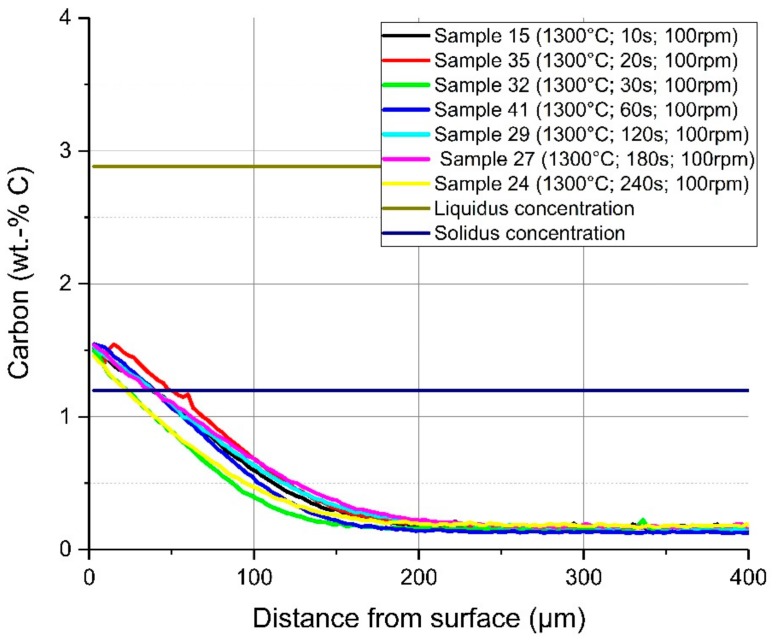
Average carbon composition and diffusion depth from the detected solid–liquid interface under forced convection in the melt through rotation of the sample at 1300 °C equilibrium temperature.

**Figure 16 materials-12-01358-f016:**
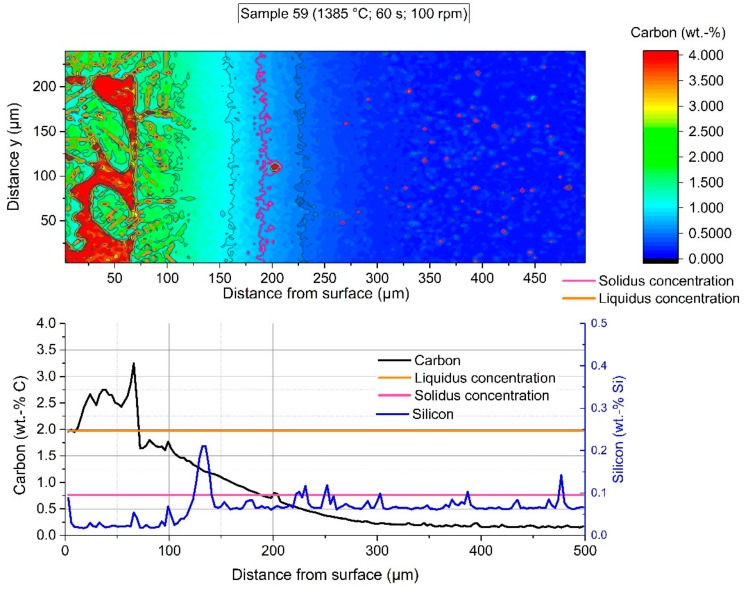
EPMA-measured carbon distribution of sample 59 with an immersion time of 60 s at 1385 °C equilibrium temperature and turbulent bath conditions.

**Figure 17 materials-12-01358-f017:**
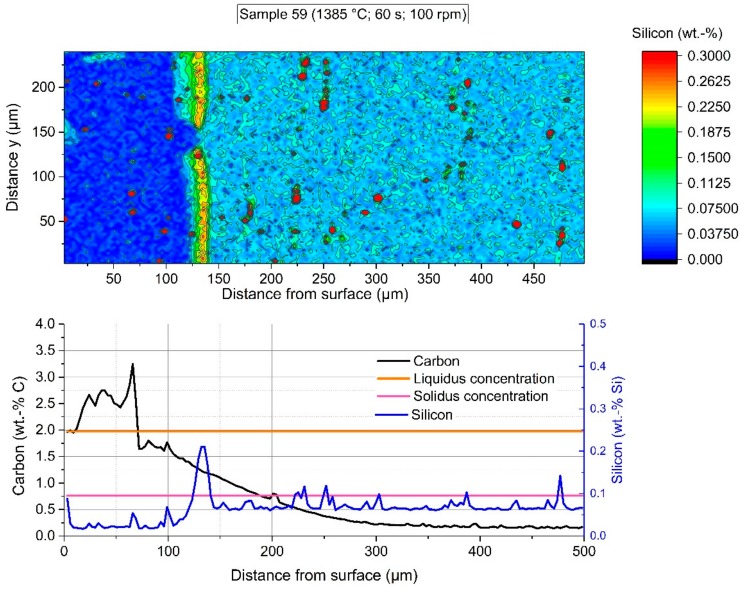
EPMA-measured silicon distribution of sample 59 with an immersion time of 60 s at 1385 °C equilibrium temperature and turbulent bath conditions.

**Table 1 materials-12-01358-t001:** Experimental data [20].

Definition	Hot Metal	Scrap
Carbon content (wt.-%)	4.58	0.1
Silicon content (wt.-%)	0.37	0.0733
Manganese content (wt.-%)	0.63	0.479
Phosphorus content (wt.-%)	0.07	0.01
Mass (g)	330	26.3
Initial temperature (°C)	1305/1370/1450	25
Equilibrium temperature (°C)	1230/1300/1385	1230/1300/1385

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
