# Peer review of "Diffusive Steel Scrap Melting in Carbon-Saturated Hot Metal—Phenomenological Investigation at the Solid–Liquid Interface"

_materials, 2019, doi:10.3390/ma12081358_

Reviewer 1 Report

There is a very well written manuscript describing quality work. It will be interesting to find out how the silicon layer was created. How would that much SiO2 have been in the scrap in the first place? I'm not sure what a 'convincible' mass transfer is (line 477). It would be nice if the conclusions suggested how these results might be used to improve scrap melting under LD converter conditions.

Author Response

Leoben, April 22nd, 2019.

Dear Reviewer 1,

We are very thankful for your review and your suggestions to our Manuscript (materials-489846) as well as your honourable statement.

According to your review we changed your points like listed below. The revised sections of your considerations are highlighted in the revised manuscript in red. Changes due to the review of reviewer 2 are highlighted in yellow.

Point 1: “There is a very well written manuscript describing quality work. It will be interesting to find out how the silicon layer was created. How would that much SiO2 have been in the scrap in the first place?”

Response: The used steel is from minor quality and has a lot of Non-metallic inclusions (silicondioxides) inside which may remain at the liquid-solid interface. Nevertheless, the Si-layer is on the scrap side so this theory might be connected with some diffusive phenomena. Further it might be possible that some counteraction between the dissolved silicon, dissolved oxygen and carbon will occur. This will be hopefully investigated in further research and will be given as a still open question to be solved in future, which was already mentioned in the conclusions. (In the revised version between line 564 and 568.)

Point 2: “I'm not sure what a 'convincible' mass transfer is (line 477).”

Response: The sentence was restructured and we hope that it is now more meaningful what we want to report.

Point 3: “It would be nice if the conclusions suggested how these results might be used to improve scrap melting under LD converter conditions.”

Response: We included in the conclusions a suggestion how the detected mathematical expression of equation 13 can be included into a dynamic LD process model. (In the revised version between line 557 and 563.)

We hope that you will be satisfied with our corrections so that the paper may be accepted and published in the Journal Materials. If you have any further doubts, please let us know and take the guarantee that we are always open for further discussions.

In the name of my co-authors and in my name, we want to thank you for your efforts!

With best regards from Austria,

Florian Markus Penz

Reviewer 2 Report

Dear Authors,

your paper is really of optimum quality and it is worthy for pubblication.

I have some minor remarks for you

1) Please separate Fig. 11 and 12 and provide pictures with higher quality.

2) at Pag. 16, please correct the caption of the Fig. 14 and 15 (now 1 and 2). And please, modify as suggested in comment 1)

Best Regards    

Author Response

Leoben, April 22nd, 2019.

 Dear Reviewer 2,

We are very thankful for your review and your suggestions to our Manuscript (materials-489846) as well as your honourable statement.

According to your review we changed your points like listed below. The revised sections of your considerations are highlighted in the revised manuscript in yellow. Changes due to the review of reviewer 1 are highlighted in red.

Point 1: “Please separate Fig. 11 and 12 and provide pictures with higher quality.”

Response: We separated the two figures. The quality of the Figures is in 1200dpi

Point 2: “at Pag. 16, please correct the caption of the Fig. 14 and 15 (now 1 and 2). And please, modify as suggested in comment 1)”

Response: The caption should be 14 and 15. I don´t know why it has changed. Hopefully it will work now with the separation of the two figures. The quality of the Figures is in 1200dpi.

We hope that you will be satisfied with our corrections so that the paper may be accepted and published in the Journal Materials. If you have any further doubts, please let us know and take the guarantee that we are always open for further discussions.

In the name of my co-authors and in my name, we want to thank you for your efforts!

With best regards from Austria,

Florian Markus Penz
